# Imaging-based assessment of muscles and malnutrition predict prognosis in patients with primary hepatocellular carcinoma

**Hitomi Takada**[ID]*, **Leona Osawa, Yasuyuki Komiyama, Masaru Muraoka, Yuichiro Suzuki, Mitsuaki Sato, Shoji Kobayashi, Takashi Yoshida, Shinichi Takano, Shinya Maekawa**[ID], **Nobuyuki Enomoto**

Gastroenterology and Hepatology Department of Internal Medicine, Faculty of Medicine, University of Yamanashi, Yamanashi, Japan

* takadahi0107@gmail.com

## Abstract

### Background

The significance of imaging-based assessment of muscles and malnutrition in patients with primary hepatocellular carcinoma (HCC) remains unclear. This study aimed to elucidate the prognostic role of the combination of Low Muscle Volume and Value (LMVV) and malnutrition.

### Methods

A total of 714 Child-Pugh grade A/ B patients with first-diagnosed HCC were enrolled, and analyzed factors associated with overall survival. LMVV was defined using psoas muscle mass index and computed tomography values of multifidus muscle at the level of the third lumbar vertebra. We used hypoalbuminemia, Child-Pugh grade B, Subjective Global Assessment (SGA) grade B/C, and Royal Free Hospital Nutrition Prioritizing Tool (RFH-NPT) score >2 as malnutrition factors in this study.

### Results

At baseline, 29% showed LMVV, and 59% met one or more of the malnutrition criteria. No items meeting the criteria of LMVV and malnutrition was observed in 41%, 1 of them was found in 29%, and both were found in 29%. The number of items meeting criteria was an independent factor for a shorter survival. The frequency of liver-related deaths did not differ by presence of LMVV alone, while it was associated with malnutrition. In contrast, the incidence of other types of deaths was influenced by LMVV and malnutrition.

**Data availability statement:** All relevant data are within the manuscript.

**Funding:** The author(s) received no specific funding for this work.

**Competing interests:** The authors have declared that no competing interests exist.

## Conclusions

The combination of LMVV and malnutrition is a prognostic factor in patients with primary HCC.

## Introduction

Sarcopenia is characterized by a low skeletal muscle mass, its weakness, and decreased physical performance [1]. It is a poor prognostic factor in patients with chronic liver disease (CLD) [2–6]. Therefore, sarcopenia has received increasing attention in recent years. According to the assessment criteria for sarcopenia in liver disease established by Japan Society of Hepatology (JSH), patients with CLD who exhibit low grip strength (GS) and muscle mass found by computed tomography (CT) or bioelectrical impedance analysis are considered to have sarcopenia [7].

Recently, there has been a growing emphasis on evaluating sarcopenia in terms of muscle quality, especially with a focus on decreased GS and elevated intramuscular fat mass [8,9]. However, it is difficult for doctors in real clinical practice to measure GS during consultation times and for dietitians to measure it during a short time of nutritional guidance. In contrast, previous studies have demonstrated the correlation between CT values of skeletal muscles at the level of the third lumbar vertebra and triglyceride content in muscle biopsy specimens and the association between CT values of skeletal muscles and prognosis in patients with unresectable hepatocellular carcinoma (HCC) [9–11]. Since muscle volume and intramuscular fat masses are determined mechanically, the inclusion of a diagnostic imaging report would eliminate the aforementioned inconvenience. Thus, imaging-based muscle assessment can be useful in busy clinical settings, but there is limited knowledge on its significance.

The evidence-based clinical practice guidelines for liver cirrhosis (LC) from the Japanese Society of Gastroenterology (JSGE)/ JSH published in 2020 reflect three criteria for initiating nutritional therapy: hypoalbuminemia (serum albumin < 3.5 g/dL), Child–Pugh grade B or C, or sarcopenia [12,13]. If patients meet even one of those of malnutrition, nutritional intervention is recommended to alleviate the condition in patients with cirrhosis [14]. Moreover, the Royal Free Hospital Nutrition Prioritizing Tool (RFH-NPT) and the Subjective Global Assessment (SGA) are useful indicators for assessing nutritional status in patients with CLD [15–17].

However, there is no report on the combination of imaging-based assessment of skeletal muscle and malnutrition, and prognosis. The significance of these criteria in patients with HCC, especially first-diagnosed, remains unclear [3,18]. Therefore, this study aimed to clarify the significance of imaging-based assessment of skeletal muscle and malnutrition assessment in patients with first-diagnosed HCC.

## Methods

### Patients

From January 2008 to January 2021, 714 patients first-diagnosed with primary HCC were enrolled in our study. The cases with adequate imaging at baseline, over

20 years old, Child–Pugh grade A or B, and performance status 0–2 were included, while those from which data could not be extracted precisely, those with the best supportive care at the time of a diagnosis of HCC, presenting Child–Pugh grade C, or those where PT activity could not be measured for the reasons, such as medication, were excluded (Fig 1).

In this study, the nodules with pathological findings, or non-rim hyper enhancement in the arterial phase of dynamic CT or gadolinium ethoxybenzyl diethylenetriamine penta-acetic acid-contrast-enhanced magnetic resonance imaging and non-peripheral washout or threshold growth that is, only when nodules that showed LR-4/5 using Liver Imaging Reporting and Data System (LI-RADS) were diagnosed as HCC [19]. Cirrhosis was defined as a platelet count less than 150,000/μL, Type IV collagen 7S of 4.4 mg/dL or greater, or hyaluronic acid of 50 ng/mL or more [20,21]. Overall survival (OS) was outlined as the time between the time of diagnosis of primary HCC and the date of death or that of last observation.

This study was approved by the Human Ethics Review Committee of Yamanashi University Hospital (approval number: 1326) in accordance with the Declaration of Helsinki. Written informed consent was obtained from all the enrolled patients. We accessed medical records for research purpose from January 2024 to June 2024. No one had access to information that could identify individual participants.

## Diagnostics of low muscle volume and value (LMVV)

According to the assessment criteria for sarcopenia in liver disease set by the JSH, the presence of decreased GS and low muscle mass in patients with CLD is indicative of sarcopenia. However, adequate GS measurements at baseline were not available due to the retrospective nature of this study. Therefore, we defined Low Muscle Volume and CT Value (LMVV) using CT scan imaging.

Psoas muscle mass index (PMI) at the level of the third lumbar vertebra, as determined by CT scan imaging, was used to assess changes in muscle mass volume. CT scan images taken at the time of the primary HCC diagnosis served as the baseline. The cross-sectional areas of bilateral psoas muscles were measured by manual tracing, and PMI was calculated by normalizing these areas to the square of a patient's height in meters. The cutoff value for PMI was identified as 6.36 and 3.92 $cm^2/m^2$ for men and women, respectively, according to the assessment criteria established by JSH [22]. CT values of multifidus muscle at the level of the third lumbar vertebra were applied to estimate skeletal muscle quality [23–25]. The cutoff value for low CT values was outlined as 44.4 and 39.3 Hounsfield unit for men and women, respectively [8,9]. In this study, LMVV was defined as the presence of both low PMI and low CT values. The measurement of muscle mass volume and CT values was conducted by two hepatology specialists with the expertise in this field.

## Diagnostics of malnutrition

Using the criteria made for initiating a nutritional therapy algorithm in the LC guideline, hypoalbuminemia or Child-Pugh grade B or C, was defined as a malnutrition factor other than sarcopenia [12,13]. Moreover, RFH-NPT score >2 and SGA

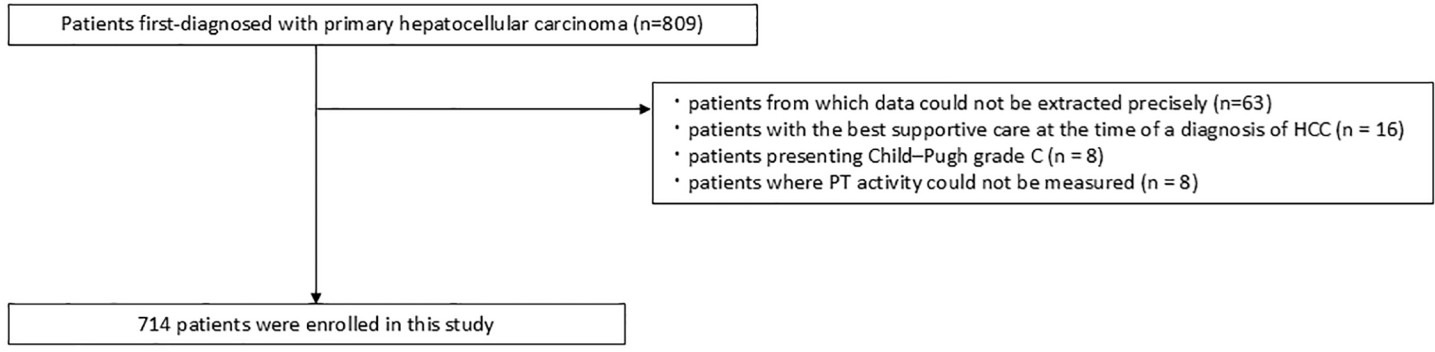

**Fig 1. Flowchart of patient enrollment.**

grade B/C were often used as indicators of malnutrition. Therefore, we used hypoalbuminemia (serum albumin < 3.5 g/dL), Child-Pugh grade B, SGA grade B/C, and RFH-NPT score > 2 as malnutrition factors in this study.

## Statistical analysis

All the obtained experimental data were expressed as medians (ranges). Between-group comparisons were conducted using Mann–Whitney, Kruskal–Wallis, Friedman tests, and non-parametric analysis of variance. If a one-way analysis of variance yielded significant results, the differences between individual groups were assessed using Fisher's exact test. The receiver operating characteristic analysis was performed, and the optimal cutoff values were determined by Youden's index. Kaplan–Meier method was used to determine OS, with the log-rank test used for the analysis. A $p$ value of < 0.05 was considered statistically significant. All statistical analyses were performed using EZR (Saitama Medical Center, Jichi Medical University, Saitama, Japan), a graphical user interface for R (The R Foundation for Statistical Computing, Vienna, Austria) [26].

## Results

### Patient characteristics

Table 1 summarizes the features of the recruited patients. A total of 714 patients were enrolled in this study. Their median age was 72 years (range, 21–93 years), and 514 were men and 200 were women. The etiologies of HCC were hepatitis B virus infection in 72 cases, hepatitis C virus infection in 402, and non-B and non-C in 240. In terms of Child–Pugh grading, 553 cases had grade A, and 161 had grade B. Regarding tumor-node-metastasis (TNM) staging, 204, 290, 159, and 61 cases were classified as stages 1, 2, 3, and 4, respectively. The maximum intrahepatic tumor diameter was 23 mm (range, 6–200 mm), and the number of them was 1 (range, 1–10). Portal venous invasion was observed in 57 cases, and distant metastasis in 46 cases. The treatment modalities for primary HCC included resection alone or in combination, percutaneous

**Table 1. Patient characteristics.**

| | Patients diagnosed with primary HCC (n = 714) |
|---|---|
| Age, years old | 72 (21-93) |
| Male, n | 514 (72%) |
| Body mass index | 23 (14-40) |
| PMI < men 6.36, women 3.92 cm$^2$/m$^2$ | 296 (42%) |
| CT value < men 44.4, women 39.3 HU | 503 (70%) |
| LMVV at baseline, n | 209 (28%) |
| Etiology (HBV/HCV/nonBnonC), n | 72/402/240 (10/56/34%) |
| Child-Pugh grade (A/B), n | 553/161 (78/22%) |
| Albumin < 3.5g/dl, n | 247 (35%) |
| SGA grade (A/B/C), n | 443/232/39 (62/33/5.5%) |
| RFH-NPT score (0/1/2<), n | 289/245/170 (41/34/24%) |
| Alpha-fetoprotein, ng/ml | 11 (1.0-623027) |
| Des-γ-carboxy prothrombin, mAU/ml | 31 (5.0-636373) |
| Tumor size, maximum, mm | 23 (6-200) |
| The number of intrahepatic tumors, n | 1 (1-10) |
| Portal vein invasion, n | 57 (8.0%) |
| Extrahepatic metastasis, n | 46 (6.4%) |
| TNM stage (1/2/3/4), n | 204/290/159/61 (29/41/22/8.0%) |
| Therapy for primary HCC (resection/ percutaneous puncture treatment/ transarterial chemoembolization/ systemic therapy/ radiotherapy, n | 224/280/172/13/25 (31/39/24/1.8/3.5%) |

puncture treatment alone or in combination, transarterial chemoembolization (TACE) alone or in combination, systemic alone or combinative therapy, and radiotherapy in 224, 280, 172, 13, and 25 cases, respectively. At baseline, 209 patients (29%) were diagnosed with LMVV. The median observation period in the study was 57 months (range, 51–64 months).

## Association between LMVV, malnutrition factors and overall survival rate

Among patients with primary HCC, those with LMVV at baseline had a poorer prognosis than those without (p = 0.047) (Fig 2a). Patients with hypoalbuminemia, Child-Pugh grade B, SGA grade B/C, RFH-NPT score >2, and meeting one or more of the malnutrition criteria had a less promising prognosis than those without (p < 0.001) (Fig – f).

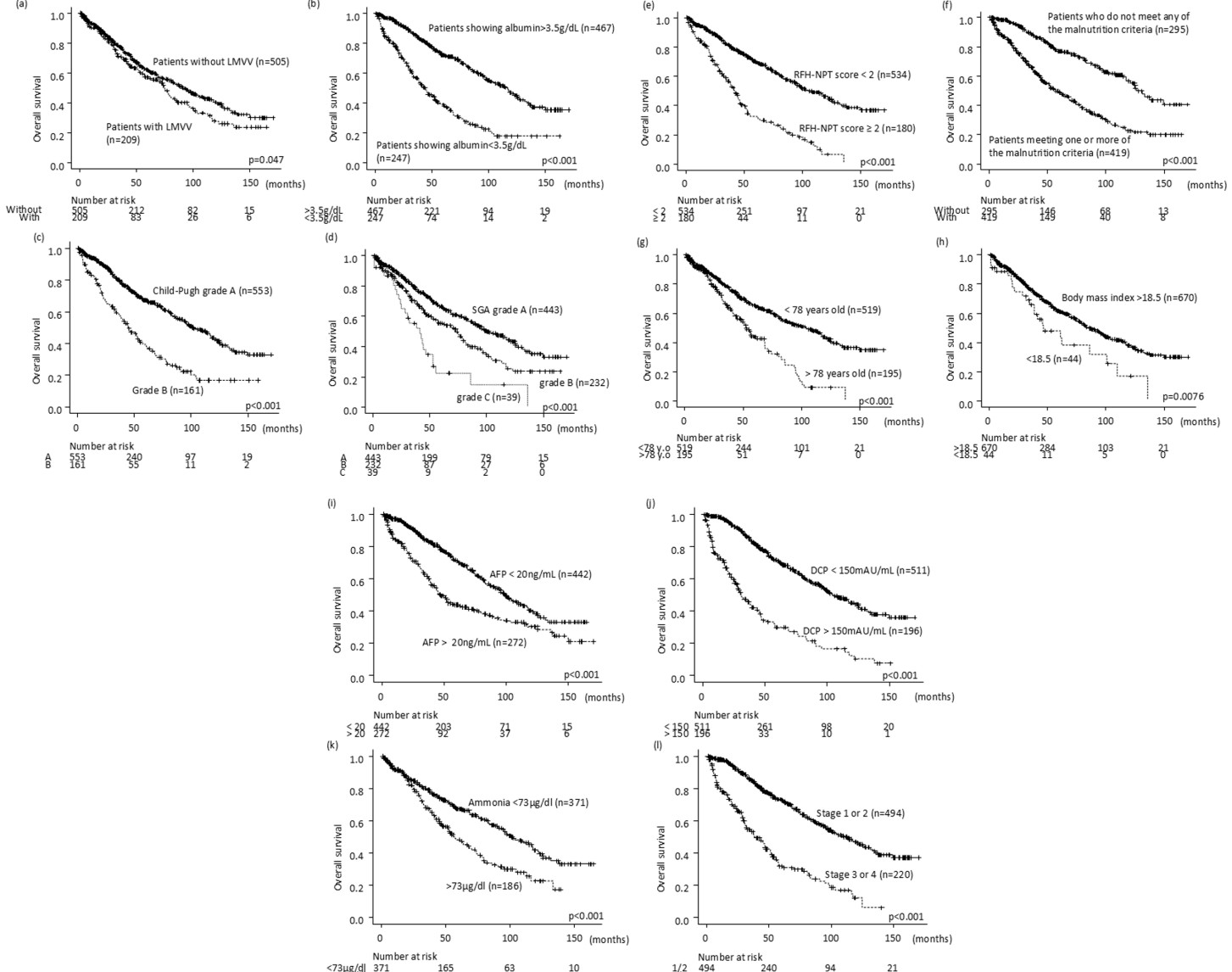

**Fig 2. Overall survival stratified by a) low muscle volume and value (LMVV), b) hypoalbuminemia, c) Child–Pugh grade B, d) Subjective Global Assessment (SGA) grade B/C, e) Royal Free Hospital Nutrition Prioritizing Tool (RFH-NPT) score ≥2, f) meeting one or more of the malnutrition criteria, g) age over 78 years, h) body mass index < 18.5, i) alpha-fetoprotein levels >20 ng/mL, j) des-μ-carboxy prothrombin levels >150 mAU/mL, k) ammonia >73μg/dl, and l) TNM stages 3 or 4.**

## Prognostic factors in patients with primary hepatocellular carcinoma

Several independent risk factors, such as age over 78 years, body mass index (BMI) < 18.5, LMVV, malnutrition, alpha-fetoprotein (AFP) levels >20 ng/mL, and des-γ-carboxy prothrombin (DCP) levels >150 mAU/mL, Ammonia > 73 µg/dL, and TNM stages 3/4 were identified for a shorter survival time in patients with primary HCC (Table 2, Fig 2g–l). The area under the curve for the cutoff value of age, BMI, AFP, DCP and ammonia levels, using operating curve analysis was 0.62 (0.58–0.66), 0.60 (0.53–0.61), 0.60 (0.56–0.64), 0.71 (0.67–0.75), and 0.57 (0.52–0.62), sensitivity 0.66, 0.30, 0.87, 0.79 and 0.43, specificity 0.54, 0.81, 0.33, 0.55 and 0.71.

## Association between LMVV and malnutrition, and overall survival rate

No parameters were found that met the criteria of LMVV and malnutrition, were observed in 295 cases (41%), 1 of them was found in 210 patients (29%), and both were found in 209 cases (29%). The prognosis was stratified by the number of items meeting criteria, and this was an independent factor for a shorter survival time (Fig 3a, Table 2). It was possible to classify the prognosis similarly for all the stages of HCC (Fig 3b–d).

## Association between LMVV, malnutrition, and liver-related/other deaths

Of the 301 deaths during the observation period, 213 (71%) were classified as liver-related (HCC or liver failure, such as hepatic encephalopathy, ascites, and bleeding). The causes of death included HCC in 136 cases, liver failure in 77, infection in 19, cancers in other organs in 16, a cardiovascular disease in 6, a cerebrovascular disease in 4, fractures in 2, and other or unknown causes in 41. For liver-related and other types of deaths, the prognosis could be delineating by the number of items answering the criteria of LMVV and malnutrition (Fig 4a,b). Meanwhile, the frequency of liver-related

**Table 2. Prognostic factors for shorter survival in patients with primary hepatocellular carcinoma.**

| | Univariate | | Multivariate Model 1 | | Multivariate Model 2 | |
|---|---|---|---|---|---|---|
| | Hazard ratio | p value | Hazard ratio | p value | Hazard ratio | p value |
| Age > 78 years old | 2.0 (1.5-2.5) | <0.001 | 1.7 (1.3-2.4) | <0.001 | 1.7 (1.3-2.3) | <0.001 |
| Male | 1.1 (0.87-1.4) | 0.40 | | | | |
| Body mass index < 18.5 | 1.8 (1.2-2.7) | 0.0059 | 1.9 (1.1-3.2) | 0.025 | 1.9 (1.1-3.2) | 0.024 |
| LMVV at baseline, n | 1.3 (1.003-1.6) | 0.047 | 1.3 (0.93-1.6) | 0.0058 | | |
| Etiology: nonBnonC | 1.5 (1.2-1.9) | <0.001 | 0.88 (0.66-1.2) | 0.40 | 0.89 (0.66-1.2) | 0.42 |
| Malnutrition | 2.7 (2.1-3.5) | <0.001 | 2.3 (1.6-3.2) | <0.001 | | |
| Child–Pugh grades B | 2.5 (2.0-3.2) | <0.001 | | | | |
| Albumin <3.5g/dL | 2.8 (2.3-3.6) | <0.001 | | | | |
| SGA Grade B or C | 1.6 (1.3-2.1) | <0.001 | | | | |
| RFH-NPT score >2 | 3.0 (2.3-3.8) | <0.001 | | | | |
| Alpha-fetoprotein >20 ng/mL | 2.0 (1.6-2.5) | <0.001 | 1.3 (1.02-1.8) | 0.038 | 1.3 (1.02-1.8) | 0.035 |
| Des-γ-carboxy prothrombin >150 mAU/mL | 3.9 (3.1-4.9) | <0.001 | 3.1 (2.3-4.1) | <0.001 | 3.1 (2.3-4.1) | <0.001 |
| Ammonia > 73 µg/dL | 1.7 (1.4-2.3) | <0.001 | 1.6 (1.2-2.1) | 0.0017 | 1.6 (1.2-2.1) | 0.0012 |
| TNM Stages 3/4 | 2.4 (2.1-2.7) | <0.001 | 2.7 (2.0-3.6) | <0.001 | 2.7 (2.0-3.6) | <0.001 |
| Tumor size, maximum, mm | 1.02 (1.02-1.03) | <0.001 | | | | |
| The number of intrahepatic tumors, n | 1.3 (1.3-1.4) | <0.001 | | | | |
| Portal vein invasion, n | 6.8 (4.9-9.5) | <0.001 | | | | |
| Extrahepatic metastasis, n | 5.5 (3.9-7.6) | <0.001 | | | | |
| Malnutrition with LMVV | | | | | 2.2 (1.6-3.0) | <0.001 |

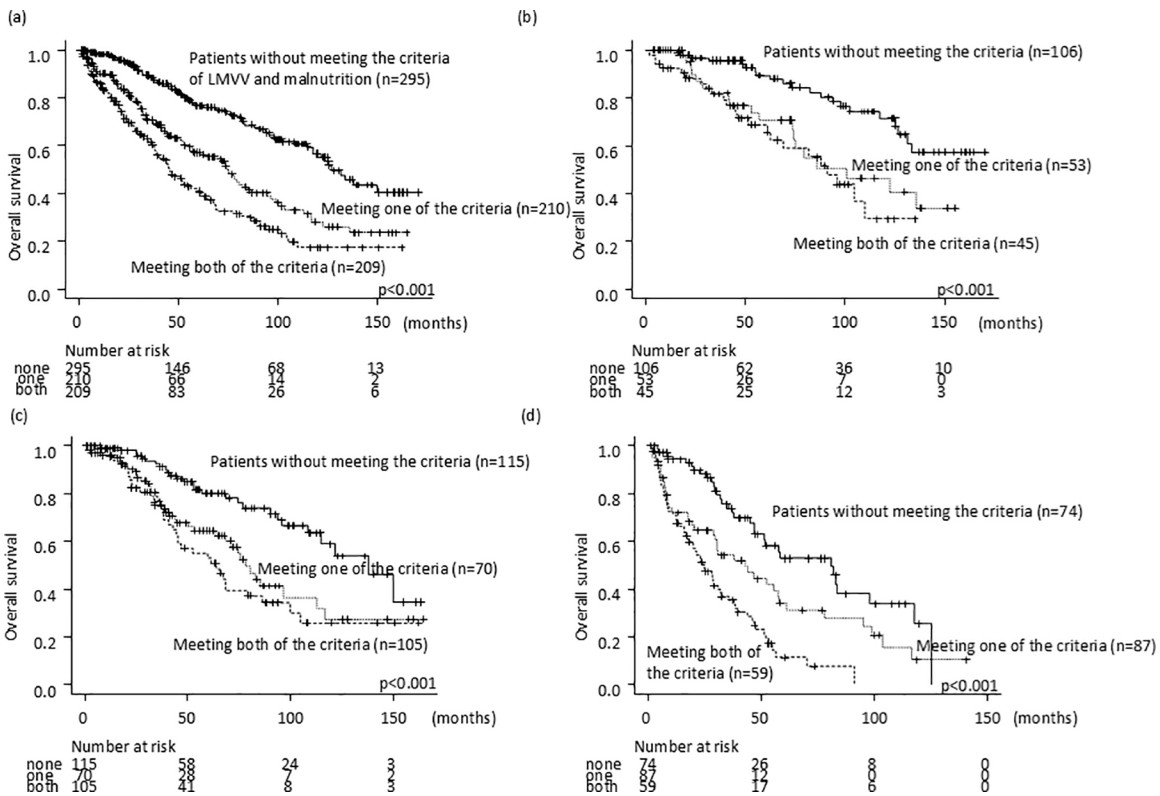

**Fig 3. Overall survival stratified by the number of items meeting the criteria of LMVV and malnutrition in a) all the patients, b) TNM stage 1, c) stage 2, and d) stage 3 or 4.**

deaths was not influenced by the presence of LMVV alone, while it was connected with malnutrition (Fig 4c,d). The incidence of other types of deaths was influenced by LMVV and malnutrition (Fig 4e,f).

## Discussion

In this study, we assessed the significance of LMVV and malnutrition in patients with first-diagnosed HCC. Our findings revealed that patients with LMVV and malnutrition at baseline had a lower OS rate than those without, regardless of the HCC stage. LMVV and malnutrition at baseline was identified as an independent prognostic factor for patients with primary HCC.

The diagnosis of sarcopenia was first proposed in 1989. Since then, several criteria have been developed [1,7,27,28]. According to the assessment criteria of JSH for sarcopenia in liver disease, low GS and muscle mass are the diagnostic features of sarcopenia. However, adequate GS measurements at baseline were not available due to the retrospective nature of this study, and we defined LMVV using only CT imaging. That is, CT values were applied instead of GS as an indicator of skeletal muscle quality in this study. CT values have already been proved to reflect muscle quality and to be strong prognostic factors [9–11]. The ability to assess muscle status in a short time with only CT imaging, without including time-consuming parameters, such as GS or gait speed, is expected to lower barriers to the assessment of muscle in patients with HCC.

We confirmed that patients with LMVV at baseline had a poorer prognosis than those without. Many recent studies have investigated the correlation between sarcopenia and prognosis in patients with HCC [3,4,6,29]. However, few of

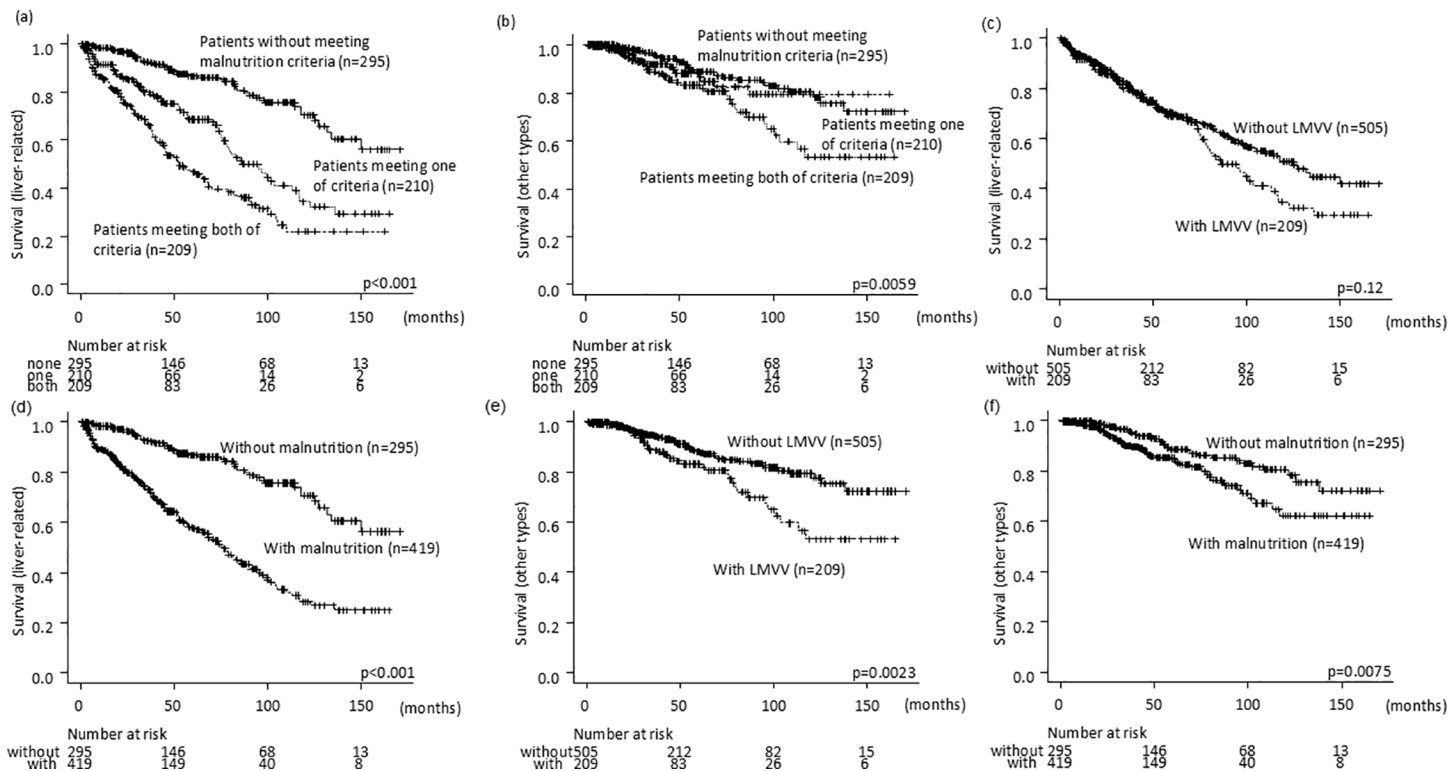

**Fig 4. Overall survival stratified by the number of items meeting the criteria of LMVV and malnutrition. a) liver-related, and b) other types of deaths. The frequency of liver-related deaths stratified by c) LMVV, and d) malnutrition. The frequency of other types of deaths stratified by e) LMVV, and f) malnutrition.**

them have examined a large number of cases with primarily diagnosed HCC over a long period of time. In addition, only few studies have classified causes of death as liver-related and other, and observed the relationship with assessment of skeletal muscle [5]. This study is only an assessment of LMVV, although it is not possible to assess the significance of sarcopenia. However, there is no report using long-term observation in such a large number of cases.

Malnutrition is defined as a measurable change in physical and mental functioning secondary to altered body composition and cell mass [30]. Malnutrition in patients with cirrhosis is reported to be connected with an increased rate of hospitalizations and deaths. It is a debatable topic that has gained attention in recent years along with sarcopenia [31]. In 2016, the Global Leadership Initiative on Malnutrition endorsed by the European Society for Clinical Nutrition and Metabolism (ESPEN) displayed a common diagnostic assessment guideline of malnutrition. ESPEN guidelines integrate sarcopenia as a major criterion for the diagnostics of malnutrition [32]. Moreover, the third edition of the LC guideline by JSGE and JSH defined three criteria of malnutrition for initiating nutritional therapy [12,13]. Therefore, this study aimed to elucidate the significance of malnutrition in patients with primary HCC, particularly in combination with skeletal muscle assessment.

We detected that the number of items meeting the criteria of LMVV and malnutrition served as an independent factor for a shorter survival time in all patients and at each stage as well. Meanwhile, the prevalence of liver-related deaths was affected by malnutrition, not LMVV. The incidence of other types of death was associated with LMVV and malnutrition. The present results revealing the link between skeletal muscle and other types of death, such as infections and fractures, have been described in patients with CLD and other diseases, and this evidence is considered reasonable [33–40].

There has been an increasing number of reports on the usefulness of the score of malnutrition, establishing the link between sarcopenia and liver function [41]. Model of end-stage liver disease (MELD) - sarcopenia score is considered an indicator of mortality in patients with CLD and readmission in patients with hepatic encephalopathy [42–44]. The albumin bilirubin (ALBI) - sarcopenia score has shown superior effectiveness in comparison with MELD-sarcopenia in patients with HCC [45]. Surprisingly, there are few reports of combined a comprehensive assessment of malnutrition with image-based assessment of skeletal muscle [14,46]. Therefore, to our knowledge, this study is the first to demonstrate that the combination in a large number of patients with primary HCC has a significant impact on a prognosis.

This study has some limitations. First, this was a single-facility retrospective study and lacked GS measurements. Therefore, it is a future task to compare the results of our research with GS measurements, which are required by the currently widely used guidelines for diagnosing sarcopenia in Japan. Second, further studies are needed to carry out a comparative analysis according to gender, etiology, and medication history. In particular, it remains unclear on which therapeutic schemes, such as branched-chain amino acids, carnitine, lactulose, and rifaximin, should be used in patients with malnutrition or sarcopenia [47–49]. The same applies to a history of nutritional guidance and exercise, which requires further investigations.

The findings of this study emphasize the significance of LMVV and malnutrition as prognostic factors in patients with HCC. The assessment of skeletal muscle using only CT imaging and the assessment of malnutrition that can be assessed using only a simple medical interview and physical measurements may easily predict the risk of liver-related and other types of mortality.

## Conclusion

LMVV and malnutrition are prognostic factors in patients with first-diagnosed primary HCC. This may create the opportunities for novel therapeutic approaches and improved outcomes. Although clinical settings focus on the treatment of HCC itself, this study shows that it is also important to assess skeletal muscle status using CT scan imaging and malnutrition in patients with HCC.

## Acknowledgements

We thank all the patients and doctors for participating in this survey.

## Author contributions

**Conceptualization:** Hitomi Takada.

**Data curation:** Hitomi Takada, Leona Osawa, Yasuyuki Komiyama, Masaru Muraoka, Yuichiro Suzuki, Mitsuaki Sato, Shoji Kobayashi, Takashi Yoshida, Shinichi Takano, Shinya Maekawa, Nobuyuki Enomoto.

**Formal analysis:** Hitomi Takada.

**Investigation:** Hitomi Takada.

**Methodology:** Hitomi Takada.

**Software:** Hitomi Takada.

**Supervision:** Hitomi Takada.

**Validation:** Hitomi Takada.

**Visualization:** Hitomi Takada.

**Writing – original draft:** Hitomi Takada.

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
