## [Decision Letter · Decision Letter 0]

1 Sep 2024

PONE-D-24-25647Malnutrition as defined by imaging-based sarcopenia and albumin levels predicts prognosis in patients with primary hepatocellular carcinomaPLOS ONE

Dear Dr. Takada,

Thank you for submitting your manuscript to PLOS ONE. After careful consideration, we feel that it has merit but does not fully meet PLOS ONE’s publication criteria as it currently stands. Therefore, we invite you to submit a revised version of the manuscript that addresses the points raised during the review process.

We look forward to receiving your revised manuscript.

Kind regards,

Nguyen Hai Nam, MD, PhD

Academic Editor

PLOS ONE

Journal Requirements:

Additional Editor Comments:

In this study, Takada et al. investigated the prognosis of HCC patients in regard to the imaging-based sarcopenia and albumin levels. Sarcopenia and malnutrition are interesting in liver disease especially in HCC but require detailed and complicated criteria. A major revision is needed to improve the completion of this manuscript as below

- At first, according to the thee universal system of sarcopenia definitions (he European Working Group on Sarcopenia in Older People (EWGSOP), the European Society for Clinical Nutrition and Metabolism Special Interest Groups (ESPEN-SIG), and the International Working Group on Sarcopenia (IWGS)), not only low skeletal muscle mass but low muscle strength or low muscle performance are strictly required to establish the presence of sarcopenia. I suggest the addition of the aforementioned elements to complete the diagnosis of sarcopenia

- On the other hand, a classification system is needed to determine the presence and the stratification of malnutrition in this study. The Royal Free Hospital Nutrition Prioritizing Tool (RFH-NPT) or the Subjective Global Assessment might be feasible and convenient in this circumstance.

- In HCC patients, the staging system is critically important in the prediction of prognosis. Therefore, the targeted population focusing on a specific group of patient, such as HCC patients with BCLC A who underwent liver resection, would convince readers. So please specific the targeted patients in this study.

- The tables were not included

Reviewers' comments:

Reviewer's Responses to Questions

**Comments to the Author**

1. Is the manuscript technically sound, and do the data support the conclusions?

Reviewer #1: Yes

Reviewer #2: Yes

2. Has the statistical analysis been performed appropriately and rigorously? 

Reviewer #1: Yes

Reviewer #2: Yes

3. Have the authors made all data underlying the findings in their manuscript fully available?

Reviewer #1: Yes

Reviewer #2: Yes

4. Is the manuscript presented in an intelligible fashion and written in standard English?

Reviewer #1: Yes

Reviewer #2: Yes

5. Review Comments to the Author

Reviewer #1: The authors investigated elucidate the prognostic role of malnutrition criteria as defined by imaging-based sarcopenia and albumin levels. The topic is of interest. Moreover, the sample size is relatively large. However, there are a few critical problems as follows.

1) Not a few investigators have so far reported the impact of sarcopenia and malnutrition on the prognosis of patients with primary HCC. Therefore, the novelty of this study is poor, although the authors combined those two parameters.

2) In the present study, the authors defined sarcopenia as low PMI and low CT values. However, the correct definition of sarcopenia is loss of skeletal muscle mass with decreased muscle strength or function. CT values are not alternatives of grip strength. So, the authors should correct sarcopenia as low skeletal muscle mass and low skeletal muscle quality throughout the article.

3) The authors used hypoalbuminemia as a parameter of malnutrition. However, in patients with liver diseases, serum albumin level is affected by liver function. Strictly speaking, therefore, it is not appropriate to use hypoalbuminemia as a marker of malnutrition in these patients.

4) In Figure 2, the label of the high AFP group in the figure is the mistake of AFP > 20 ng/mL.

Reviewer #2: In this study, Takada et al. analyzed the outcome of firstly diagnosed HCC patients focusing on the status of sarcopenia and hypoalbuminemia. The presence of sarcopenia was evaluated using CT images. The outcome was finely evaluated separately for liver-related deaths and other types of deaths. This study concluded that sarcopenia and hypoalbuminemia are prognostic factor in HCC patients. The analyses performed well but there are several issues to be addressed as below.

1. In the manuscript, tables are not included.

2. Page 5, it is described that sarcopenia was defined as the presence of both low PMI and low CT values. In the JSH guideline, low CT values of muscles are not included in the definition of sarcopenia. In general, a low-density muscle in CT is considered as myosteatosis. Why did the authors include the CT value for the definition of sarcopenia? Please describe the rationale for this definition.

3. Figure 2—the Kaplan-Meier curves were constructed using several parameters. How were the cutoff values of parameters determined?

4. Page 5, “The cutoff value for low CT values was outlined as 44.4 and 39.3 Hounsfield unit for women and women, respectively”—please fix the part “women and women”.

5. Page 7, “No parameters meeting the criteria were observed in 330 cases”—the criteria are unclear in this sentence. Please clearly show what are the criteria in the main text.

6. Figure 3—please show what are the malnutrition criteria in the figure legend.

6. PLOS authors have the option to publish the peer review history of their article (what does this mean? ). If published, this will include your full peer review and any attached files.

**Do you want your identity to be public for this peer review?** For information about this choice, including consent withdrawal, please see our Privacy Policy .

Reviewer #1: No

Reviewer #2: **Yes: ** Jun Inoue

---

## [Author Response · Author response to Decision Letter 0]

16 Oct 2024

Dear Dr. Nguyen Hai Nam,

Academic Editor

and the reviewers, PLOS ONE

PONE-D-24-25647

Malnutrition as defined by imaging-based sarcopenia and albumin levels predicts prognosis in patients with primary hepatocellular carcinoma

Thank you for your kind review of our manuscript.

We appreciate receiving this opportunity to re-submit our manuscript.

Considering the suggestions, we have written Point by Points. We believe that these changes will alleviate the concerns of the reviewers. We would greatly appreciate if our manuscript could be accepted for publication in PLOS ONE.

Sincerely,

Hitomi Takada, MD.

Gastroenterology and Hepatology Department of Internal Medicine, Faculty of Medicine,

University of Yamanashi,

Yamanashi, Japan 

Point by points

 Reviewer 1

1. Not a few investigators have so far reported the impact of sarcopenia and malnutrition on the prognosis of patients with primary HCC. Therefore, the novelty of this study is poor, although the authors combined those two parameters.

Thank you very much for your detailed comments. As you pointed out, there are enough published reports about the impact of the combination of sarcopenia and malnutrition in patients with chronic liver disease. However, there is no report on the combination and prognosis in patients with HCC, and we consider this study to be important, as it evaluated and followed a large number of patients, in particular, it only included patients with first-diagnosed HCC. In Japan, the first guideline for rehabilitation treatment for cancer was also published in 2013, but there are still no established evaluation methods and indicators for the start of rehabilitation in cancer patients worldwide. Therefore, we have considered that the present study on low muscle volume and CT value (LMVV) and malnutrition in a large number of HCC cases was useful.

We have modified the text as follows. (line 73, page 3)

Moreover, there are a few published reports about the impact of the combination of sarcopenia and malnutrition in patients with CLD. However, there is no report on the combination and prognosis in patients with HCC. The significance of these criteria in patients with HCC, especially first-diagnosed, remains unclear 3,18.

2. In the present study, the authors defined sarcopenia as low PMI and low CT values. However, the correct definition of sarcopenia is loss of skeletal muscle mass with decreased muscle strength or function. CT values are not alternatives of grip strength. So, the authors should correct sarcopenia as low skeletal muscle mass and low skeletal muscle quality throughout the article.

Thank you very much for your comments. As you pointed out, our description was inappropriate. We defined Low Muscle Volume and CT Value (LMVV) using CT scan imaging, and corrected description of sarcopenia.

We have modified the text, tables, and figures as follows.

(Title)

Imaging-based assessment of muscles and malnutrition predict prognosis in patients with primary hepatocellular carcinoma

(Short title)

Low muscle mass, low CT values, and malnutrition in patients with primary HCC

(line 27, page 2)

The significance of imaging-based assessment of muscles and malnutrition in patients with primary hepatocellular carcinoma (HCC) remains unclear. This study aimed to elucidate the prognostic role of the combination of Low Muscle Volume and Value (LMVV) and malnutrition.

LMVV was defined using psoas muscle mass index and computed tomography values of multifidus muscle at the level of the third lumbar vertebra.

The combination of LMVV and malnutrition is a prognostic factor in patients with primary HCC.

At baseline, 29% showed LMVV, and 24% showed RFH-NPT score >2. No items meeting the criteria of LMVV and RFH-NPT score >2 was observed in 54%, 1 of them was found in 37%, and both were found in 9%. The number of items meeting criteria was an independent factor for a shorter survival. The frequency of liver-related deaths did not differ by presence of LMVV alone, while it was associated with malnutrition. In contrast, the incidence of other types of deaths was influenced by LMVV and malnutrition.

The combination of LMVV and malnutrition is a prognostic factor in patients with primary HCC.

(line 46, page 2)

Keywords: skeletal muscle, malnutrition, hepatocellular carcinoma, sarcopenia

(line 65, page 3)

Imaging-based muscle assessment can be useful in busy clinical settings, but there is limited knowledge on its significance.

(line 77, page 4)

Therefore, this study aimed to clarify the significance of imaging-based assessment of skeletal muscle and malnutrition assessment in patients with first-diagnosed HCC.

(line 105, page 5)

Diagnostics of Low Muscle Volume and Value (LMVV)

Therefore, CT values were analyzed instead of GS as indicators of skeletal muscle quality, and we defined Low Muscle Volume and CT Value (LMVV) using CT scan imaging.

In this study, LMVV was defined as the presence of both low PMI and low CT values.

(line 157, page 7)

At baseline, 209 patients (29%) were diagnosed with LMVV.

(line 163, page 8)

Association Between LMVV, malnutrition factors and Overall Survival Rate

Among patients with primary HCC, those with LMVV at baseline had a poorer prognosis than those without (p = 0.047) (Figure.2a).

The prognosis was stratified when patients were categorized according to the presence of LMVV and malnutrition factors (Figure.2f-i).

(line 181, page 9)

Several independent risk factors, such as age over 78 years, LMVV, Child–Pugh grade B, RFH-NPT score >2, alpha-fetoprotein (AFP) levels >20 ng/mL, and des-γ-carboxy prothrombin (DCP) levels >150 mAU/mL, and TNM stages 3/4were identified for a shorter survival time in patients with primary HCC (Table 2, Figure.2j-m).

(line 223, page 13)

In this study, we assessed the significance of LMVV and malnutrition in patients with first-diagnosed HCC.

(line 231, page 13)

However, adequate GS measurements at baseline were not available due to the retrospective nature of this study, and we defined LMVV using only CT imaging. That is, CT values were applied instead of GS as an indicator of skeletal muscle quality in this study. CT values have already been proved to reflect muscle quality and to be strong prognostic factors 9-11. The ability to assess muscle status in a short time with only CT imaging, without including time-consuming parameters, such as GS or gait speed, is expected to lower barriers to the assessment of muscle in patients with HCC.

(line 239, page 13)

We confirmed that patients with LMVV at baseline had a poorer prognosis than those without. Many recent studies have investigated the correlation between sarcopenia and prognosis in patients with HCC 3,4,6,29. However, few of them have examined a large number of cases with primarily diagnosed HCC over a long period of time, as conducted in this study. In addition, only few studies have classified causes of death as liver-related and other, and observed the relationship with assessment of skeletal muscle.

(line 254, page 13)

Therefore, this study aimed to elucidate the significance of malnutrition in patients with primary HCC, particularly in combination with skeletal muscle assessment.

(line 266, page 15)

Meanwhile, the prevalence of liver-related deaths was affected by high RFH-NPT score, not LMVV. The incidence of other types of death was associated with LMVV and high RFH-NPT score. The present results revealing the link between skeletal muscle and other types of death, such as infections and fractures, have been described in patients with CLD and other diseases, and this evidence is considered reasonable 33-40.

(line 277, page 15)

Surprisingly, there are few reports of combined RFH-NPT score and imaging-based assessment of skeletal muscle 14,46.

(line 291, page 15)

The findings of this study emphasize the significance of LMVV and RFH-NPT score >2 as prognostic factors in patients with HCC. The assessment of skeletal muscle using only CT imaging and the assessment of malnutrition that can be assessed using only a simple medical interview and physical measurements may easily predict the risk of liver-related and other types of mortality.

3. The authors used hypoalbuminemia as a parameter of malnutrition. However, in patients with liver diseases, serum albumin level is affected by liver function. Strictly speaking, therefore, it is not appropriate to use hypoalbuminemia as a marker of malnutrition in these patients.

Thank you very much for your comments. As you pointed out, our description was inappropriate using hypoalbuminemia as a marker of malnutrition. As markers of malnutrition, assessment using the Royal Free Hospital Nutrition Prioritizing Tool (RFH-NPT) and the Subjective Global Assessment (SGA) has been added to the revised manuscript.

We have modified the text as follows.

(line 34, page 2)

We used hypoalbuminemia, Child-Pugh grade B, Subjective Global Assessment (SGA) grade B/C, and Royal Free Hospital Nutrition Prioritizing Tool (RFH-NPT) score >2 as malnutrition factors in this study.

(line 126, page 6)

Moreover, the Royal Free Hospital Nutrition Prioritising Tool (RFH-NPT) and the Subjective Global Assessment (SGA) are useful indicators for assessing nutritional status in patients with CLD 15-17.

(line 166, page 8)

Patients with hypoalbuminemia, Child-Pugh grade B, SGA grade B/C, and RFH-NPT score >2 had a less promising prognosis than those without (p < 0.001) (Figure.2b-e). The prognosis was stratified when patients were categorized according to the presence of LMVV and malnutrition factors (Figure.2f-i).

Several independent risk factors, such as age over 78 years, LMVV, Child–Pugh grade B, RFH-NPT score >2, alpha-fetoprotein (AFP) levels >20 ng/mL, and des-γ-carboxy prothrombin (DCP) levels >150 mAU/mL, and TNM stages 3/4were identified for a shorter survival time in patients with primary HCC (Table 2, Figure.2j-m).

(line 191, page 12)

Association Between LMVV, malnutrition using RFH-NPT score, and Overall Survival Rate

No parameters were found that met meeting the criteria of LMVV and RFH-NPT score >2, were observed in 389 cases (54%), 1 of them was found in 261 patients (37%), and both were found in 64 cases (9%).

(line 210, page 12)

Association Between LMVV, RFH-NPT score, and Liver-related/Other Deaths

For liver-related and other types of deaths, the prognosis could be delineating by the number of items answering the criteria of LMVV and RFH-NPT score >2 (Figure.4a, b).

(line 224, page 13)

Our findings revealed that patients with LMVV and high RFH-NPT score at baseline had a lower OS rate than those without, regardless of the HCC stage. LMVV and high RFH-NPT score at baseline was identified as an independent prognostic factor for patients with primary HCC.

(line 258, page 14)

We detected that the number of items meeting the criteria of LMVV and RFH-NPT score >2 served as an independent factor for a shorter survival time in all patients and at each stage as well.

4. In Figure 2, the label of the high AFP group in the figure is the mistake of AFP > 20 ng/mL.

Thank you very much for your comments. As you pointed out, our description was inappropriate.

We have modified Figure 2.

 Reviewer 2

1. In the manuscript, tables are not included.

Thank you very much for your comments. We have added tables in revised manuscript.

2. Page 5, it is described that sarcopenia was defined as the presence of both low PMI and low CT values. In the JSH guideline, low CT values of muscles are not included in the definition of sarcopenia. In general, a low-density muscle in CT is considered as myosteatosis. Why did the authors include the CT value for the definition of sarcopenia? Please describe the rationale for this definition.

Thank you very much for your detailed comments. As you pointed out, our description in the original manuscript was inappropriate. However, adequate GS measurements at baseline were not available due to the retrospective nature of this study. Therefore, we defined Low Muscle Volume and CT Value (LMVV) using CT scan imaging, and corrected description of sarcopenia.

We have modified the text, tables and figures. The corrected sections are shown on page 3 of this document. Due to the length of the document, multiple mentions have been omitted.

3. Figure 2—the Kaplan-Meier curves were constructed using several parameters. How were the cutoff values of parameters determined?

Thank you very much for your comments. As you pointed out, our description in the original manuscript was inappropriate. In this study, the receiver operating characteristic analysis was performed, and the optimal cutoff values were determined by Youden’s index.

We have modified the text as follows. (line 184, page 9)

The area under the curve for the cutoff value of age, AFP and DCP levels using operating curve analysis was 0.62 (0.58-0.66), 0.60 (0.56-0.64), and 0.71 (0.67-0.75), sensitivity 0.66, 0.87 and 0.79, specificity 0.54, 0.33 and 0.55.

4. Page 5, “The cutoff value for low CT values was outlined as 44.4 and 39.3 Hounsfield unit for women and women, respectively”—please fix the part “women and women”.

Thank you very much for your kind comments. As you pointed out, our description in the original manuscript was inappropriate.

We have modified the text as follows. (line 117, page 5)

The cutoff value for low CT values was outlined as 44.4 and 39.3 Hounsfield unit for men and women, respectively 8,9.

5. Page 7, “No parameters meeting the criteria were observed in 330 cases”—the criteria are unclear in this sentence. Please clearly show what are the criteria in the main text.

Thank you very much for your comments. As you pointed out, our description in the original manuscript was unclear. In addition, the description has been corrected within the revised manuscript using the new criteria using LMVV and RFH-NPT score.

We have modified the text as follows. (line 193, page 12)

No parameters were found that met the criteria of LMVV and RFH-NPT score >2, were observed in 389 cases (54%), 1 of them was found in 261 patients (37%), and both were found in 64 cases (9%).

6. Figure 3—please show what are the malnutrition criteria in the figure legend.

Thank you very much for your kind comments. As you pointed out, our description was inappropriate.

We have modified the figure legend.

Figure 3. Overall survival stratified by the number of items meeting the criteria of LMVV and RFH-NPT score >2 in a) all the patients, b) TNM stage 1, c) stage 2, d) stage 3 or 4, e) BCLC stage A, f) stage B, and g) stage C.

 Editor comments

1. At first, according to the universal system of sarcopenia definitions (he European Working Group on Sarcopenia in Older People (EWGSOP), the European Society for Clinical Nutrition and Metabolism Special Interest Groups (ESPEN-SIG), and the International Working Group on Sarcopenia (IWGS)), not only low skeletal muscle mass but low muscle strength or low muscle performance are strictly required to establish the presence of sarcopenia. I suggest the addition of the aforementioned elements to complete the diagnosis of sarcopenia?

Thank you very much for your detailed comments. As you pointed out, our description in the original manuscript was inappropriate. However, adequate GS measurements at baseline were not available due to the retrospective nature of this study. Therefore, we defined Low Muscle Volume and CT Value (LMVV) using CT scan imaging, and corrected description of sarcopenia.

We have modified the text, tables and figures. The corrected sections are shown on page 3 of this document. Due to the length of the document, multiple mentions have been omitted.

2. On the other hand, a classification system is needed to determine the presence and the stratification of malnutrition in this study. The Royal Free Hospital Nutrition Prioritizing Tool (RFH-NPT) or the Subjective Global Assessment might be feasible and convenient in this circumstance.

Thank you very much for your comments. As you pointed out, our description was inappropriate using hypoalbuminemia as a marker of malnutrition. As markers of malnutrition, assessment using the Royal Free Hospital Nutrition Prioritizing Tool (RFH-NPT) and the Subjective Glob

---

## [Editor Report · Decision Letter 1]

25 Nov 2024

PONE-D-24-25647R1Imaging-based assessment of muscles and malnutrition predict prognosis in patients with primary hepatocellular carcinomaPLOS ONE

Dear Dr. Takada,

Thank you for submitting your manuscript to PLOS ONE. After careful consideration, we feel that it has merit but does not fully meet PLOS ONE’s publication criteria as it currently stands. Therefore, we invite you to submit a revised version of the manuscript that addresses the points raised during the review process.

We look forward to receiving your revised manuscript.

Kind regards,

Nguyen Hai Nam, MD, PhD

Academic Editor

PLOS ONE

Additional Editor Comments:

Thank you so much for your revised manuscript. Although critical adjustment have been made, there are still several points that need to be improved as below:

1. Since the criteria of sarcopenia, in this manuscript, cannot be met due to the lack of GS measurement data, the Low Muscle Volume and CT Value (LMVV) must be an appropriately alternative option. In this context, the author must edit the manuscript with the limitation of using the word “sarcopenia”

2. Regarding malnutrition, it’s reasonable to use hypoalbuminemia, Child-Pugh grade B, Subjective Global Assessment (SGA) grade B/C, and Royal Free Hospital Nutrition Prioritizing Tool (RFH-NPT) score >2 as malnutrition factors. However, I strongly suggest to use the combination of these above factors for the establishment malnutrition, instead of separate assessment of each factor.

3. Regarding BCLC staging, since the BCLC stage B group did not show similar results to the other groups, I strongly suggest to remove the patient with BCLC stage B and C; and we just focus on BCLC stage A. Please redo your analysis

---

## [Author Response · Author response to Decision Letter 1]

7 Jan 2025

Dear Dr. Nguyen Hai Nam,

Academic Editor

and the reviewers, PLOS ONE

PONE-D-24-25647

Malnutrition as defined by imaging-based sarcopenia and albumin levels predicts prognosis in patients with primary hepatocellular carcinoma

Thank you for your kind review of our manuscript.

We appreciate receiving this opportunity to re-submit our manuscript.

Considering the suggestions, we have written Point by Points. We believe that these changes will alleviate the concerns of the reviewers. We would greatly appreciate if our manuscript could be accepted for publication in PLOS ONE.

Sincerely,

Hitomi Takada, MD.

Gastroenterology and Hepatology Department of Internal Medicine, Faculty of Medicine,

University of Yamanashi,

Yamanashi, Japan 

Point by points

1. Since the criteria of sarcopenia, in this manuscript, cannot be met due to the lack of GS measurement data, the Low Muscle Volume and CT Value (LMVV) must be an appropriately alternative option. In this context, the author must edit the manuscript with the limitation of using the word “sarcopenia”.

Thank you very much for your detailed comments. As you pointed out, LMVV and sarcopenia are different concepts. We stated more carefully that this study is only an examination using LMVV, and excluded sarcopenia from keywords.

We have modified the text as follows. (line 236, page 14)

This study is only an assessment of LMVV, although it is not possible to assess the significance of sarcopenia.

2. Regarding malnutrition, it’s reasonable to use hypoalbuminemia, Child-Pugh grade B, Subjective Global Assessment (SGA) grade B/C, and Royal Free Hospital Nutrition Prioritizing Tool (RFH-NPT) score >2 as malnutrition factors. However, I strongly suggest to use the combination of these above factors for the establishment malnutrition, instead of separate assessment of each factor.

Thank you very much for your detailed comments. As you pointed out, malnutrition should be assessed comprehensively, and the text, subtitles, and figures have been revised.

(line 37 page 2)

At baseline, 29% showed LMVV, and 59% met one or more of the malnutrition criteria. No items meeting the criteria of LMVV and malnutrition was observed in 41%, 1 of them was found in 29%, and both were found in 29%.

(line 164 page 8)

Patients with hypoalbuminemia, Child-Pugh grade B, SGA grade B/C, RFH-NPT score >2, and meeting one or more of the malnutrition criteria had a less promising prognosis than those without (p < 0.001) (Figure.2b-f).

(line 176 page 9)

Several independent risk factors, such as age over 78 years, body mass index (BMI) < 18.5, LMVV, malnutrition, alpha-fetoprotein (AFP) levels >20 ng/mL, and des-γ-carboxy prothrombin (DCP) levels >150 mAU/mL, Ammonia > 73 μg/dL, and TNM stages 3/4 were identified for a shorter survival time in patients with primary HCC (Table 2, Figure.2g-l).

(line 188 page 12)

No parameters were found that met the criteria of LMVV and malnutrition, were observed in 295 cases (41%), 1 of them was found in 210 patients (29%), and both were found in 209 cases (29%).

(line 203 page 12)

For liver-related and other types of deaths, the prognosis could be delineating by the number of items answering the criteria of LMVV and malnutrition (Figure.4a, b). Meanwhile, the frequency of liver-related deaths was not influenced by the presence of LMVV alone, while it was connected with malnutrition (Figure.4c, d). The incidence of other types of deaths was influenced by LMVV and malnutrition (Figure.4e, f).

(line 216 page 13)

In this study, we assessed the significance of LMVV and malnutrition in patients with first-diagnosed HCC. Our findings revealed that patients with LMVV and malnutrition at baseline had a lower OS rate than those without, regardless of the HCC stage. LMVV and malnutrition at baseline was identified as an independent prognostic factor for patients with primary HCC.

(line 252 page 14)

We detected that the number of items meeting the criteria of LMVV and malnutrition served as an independent factor for a shorter survival time in all patients and at each stage as well. Meanwhile, the prevalence of liver-related deaths was affected by malnutrition, not LMVV. The incidence of other types of death was associated with LMVV and malnutrition.

(line 265 page 15)

Surprisingly, there are few reports of combined a comprehensive assessment of malnutrition with image-based assessment of skeletal muscle [14, 46].

(line 279 page 15)

The findings of this study emphasize the significance of LMVV and malnutrition as prognostic factors in patients with HCC.

(line 285 page 16)

LMVV and malnutrition are prognostic factors in patients with first-diagnosed primary HCC.

3. Regarding BCLC staging, since the BCLC stage B group did not show similar results to the other groups, I strongly suggest to remove the patient with BCLC stage B and C; and we just focus on BCLC stage A. Please redo your analysis.

Thank you very much for your detailed comments. As you pointed out, the failure to stratify prognosis by the combination of LMVV and malnutrition in the BCLC stage B as in other populations, is troubling, and it is an excellent way to exclude these cases and include only BCLC stage A cases.

However, the key findings of this study are that LMVV and malnutrition are important factors across patients with HCC of diverse stages, and that malnutrition and skeletal muscle assessment are important independently of TNM stages. Therefore, we wanted to use the results of all 714 cases and did not restrict this study to BCLC stage A cases and revised the text as follows.

In addition, BCLC staging is a clinically useful classification, but it is a complex classification consisting of items related to HCC staging, liver function and malnutrition. Therefore, it was the author's intention to make a simpler comparison in this study, and the relevance of BCLC staging has been removed.

---

## [Editor Report · Decision Letter 2]

26 Feb 2025

Imaging-based assessment of muscles and malnutrition predict prognosis in patients with primary hepatocellular carcinoma

PONE-D-24-25647R2

Dear Dr. Takada,

We’re pleased to inform you that your manuscript has been judged scientifically suitable for publication and will be formally accepted for publication once it meets all outstanding technical requirements.

Kind regards,

Nguyen Hai Nam, MD, PhD

Academic Editor

PLOS ONE
---

## [Editor Report · Acceptance letter]

PONE-D-24-25647R2

PLOS ONE

Dear Dr. Takada,

I'm pleased to inform you that your manuscript has been deemed suitable for publication in PLOS ONE. Congratulations! Your manuscript is now being handed over to our production team.

Kind regards,

on behalf of

Dr. Nguyen Hai Nam

Academic Editor

PLOS ONE